# Comparative Study of Distributed Acoustic Sensing Responses in Telecommunication Optical Cables

**DOI:** 10.3390/s25247600

**Published:** 2025-12-15

**Authors:** Abdulfatah A. G. Abushagur, Mohd Ridzuan Mokhtar, Noor Shafikah Md Rodzi, Khazaimatol Shima Subari, Siti Azlida Ibrahim, Zulkifli Mahmud, Zulfadzli Yusoff, Andre Franzen, Hairul Abdul Rashid

**Affiliations:** 1Centre of Fibre Networking and Communication, COE for Intelligent Network, Multimedia University, Cyberjaya 63100, Malaysia; shafikahrodzi@gmail.com (N.S.M.R.); khazaimatol@dkss.com.my (K.S.S.); azlida@mmu.edu.my (S.A.I.); zulkifli.mahmud07@gmail.com (Z.M.); zulfadzli.yusoff@mmu.edu.my (Z.Y.); hairul@mmu.edu.my (H.A.R.); 2Faculty of Artificial Intelligence and Engineering, Multimedia University, Cyberjaya 63100, Malaysia; 3Petronas Research Centre SDN BHD, Jalan Ayar Itam, Kawasan Institusi Bangi, Bandar Baru Bangi 43000, Malaysia; andref@franzen-my.com

**Keywords:** distributed acoustic sensing (DAS), optical fibre sensing, telecommunication cables, infrastructure monitoring, acoustic monitoring

## Abstract

**Highlights:**

**What are the main findings?**
The Distributed Acoustic Sensing (DAS) response is strongly influenced by the cable’s structural configuration, with the Microcable showing the highest strain sensitivity (≈0.025 a.u.) compared to the Duct and Anti-Rodent cables tested under identical laboratory conditions.Simplified and compact cable designs enable more effective strain transfer to the optical fiber, whereas multilayered or armored constructions significantly attenuate the transmitted strain signal.

**What are the implications of the main findings?**
The structural configuration of telecommunication cables should be carefully considered when repurposing them for DAS applications, as it directly impacts sensing efficiency and spatial response.Microcables are ideal for indoor or duct-protected environments requiring high sensitivity, while armored cables remain preferable for long-term or harsh-environment installations prioritizing durability.

**Abstract:**

Distributed Acoustic Sensing (DAS) transforms conventional optical fibres into large-scale acoustic sensor arrays. While existing telecommunication cables are increasingly considered for DAS-based monitoring, their performance depends strongly on cable construction and strain transfer efficiency. In this study, the relative DAS signal amplitudes of three commercial telecommunication optical cables were experimentally compared using a benchtop Rayleigh backscattering-based interrogator under controlled laboratory conditions. By maintaining a constant temperature and ensuring no additional strain changes from the outside environment, we guaranteed that only strain-induced variations from acoustic excitations were measured. The results show clear differences in signal amplitude and signal-to-noise ratio (SNR) among the tested cables. The Microcable consistently produced the highest spatial peak amplitude (up to 0.029 a.u.) and SNR (up to 79), while the Duct cable reached 0.00268 a.u. with mean SNR ≈ 32. The Anti-Rodent cable showed low signal amplitude (0.0018 a.u.) but exhibited a high mean SNR (≈111) driven by an exceptional low noise floor in one of the runs. These findings reflect the variations in mechanical coupling between the fibre core and external perturbations and provide practical insights into the suitability of different telecom cable types for DAS applications, supporting informed choices for future deployments.

## 1. Introduction

Distributed Acoustic Sensing (DAS) has emerged as a powerful technique for transforming standard optical fibres into large scale acoustic sensor arrays [1,2]. By analyzing the coherent Rayleigh backscattered light along a fibre, DAS enables the detection of dynamic strain variations induced by acoustic including fast vibration events [3,4,5]. This capability has attracted considerable interest in a wide range of applications, including seismic monitoring, perimeter security, traffic surveillance, and structural health assessment [6,7,8]. A key advantage of DAS is its compatibility with existing telecommunication fibre networks, which already span vast geographical areas and offer a cost-effective infrastructure for sensing deployment. This transformative approach repurposes idle also known as “dark” fibres as dense seismic arrays, enabling dynamic high-resolution strain measurements without the need for conventional geophones or accelerometers.

Zhan (2020) [9] demonstrated that DAS repurposes existing telecommunication fibre cables as dense seismic arrays using Rayleigh backscattering, detecting broadband seismic signals with high fidelity and metre-scale resolution, even on idle “dark” fibres. The study also discussed coupling and directional sensitivity challenges while showcasing DAS’s potential for earthquake monitoring, structural imaging, and environmental sensing. Lior et al. (2021) extended this concept to underwater environments, showing that DAS on submarine telecom cables can detect earthquakes with sensitivity comparable to broadband seismometers, particularly in deep-sea regions [10]. Fernández-Ruiz et al. (2022) [7] reviewed DAS applications across diverse environments, highlighting chirped-pulse systems for improved low-frequency sensitivity and scalability using existing telecom fibres in ocean-bottom deployments. Zhu et al. (2022) [6] emphasized DAS’s versatility for monitoring railways, tunnels, and pipelines, describing real-time, long-range, and high-density field implementations. They highlighted successful field deployments using telecom fibres and discussed installation methods, coupling challenges, and intelligent algorithms for train tracking and infrastructure health assessment. Rafi et al. (2024) compared single-mode and multimode fibres in DAS-based geophysical exploration, reporting superior strain sensitivity for single-mode fibres in vertical seismic profiling [11], while Liu et al. (2025) [12] demonstrated that pre-existing telecom networks can support city-scale DAS for real-time seismic, traffic, and structural monitoring without disrupting communication services.

Although these studies have examined strain-transfer behaviour in optical fibres and purpose-built sensing cables, very limited work has focused on the quantitative DAS performance of commercially deployed telecommunication cables, particularly those widely used in access networks such as Anti-Rodent, Duct, and Microcable designs. Previous comparisons, such as Freeland et al. (2017) [13], evaluated a mixture of specialty sensing cables and selected commercial designs which are almost identical in their design, but did not study the effect of the cable constructions nor assess their DAS responses under identical controlled excitation. Moreover, most prior work reported qualitative trends rather than providing direct amplitude-based comparisons acquired under uniform interrogator settings and a standardized mechanical coupling configuration. The present study fills this gap by offering the first systematic, side-by-side quantitative assessment of these three commonly deployed telecom cable types obtained from Dura-Mine Sdn. Bhd [14] using the same Rayleigh-based interrogator, excitation method, gauge-length configuration, and coupling conditions. This approach isolates the effect of cable architecture on DAS performance and provides practical guidance for repurposing existing fibre infrastructure for distributed sensing, an aspect of high relevance as operators increasingly explores DAS functionality without installing dedicated sensing cables.

## 2. Materials and Methods

### 2.1. Optical Cable Samples

Three types of commercial telecommunication optical cables manufactured by Dura-Mine Sdn. Bhd. were selected for this study: Anti-Rodent Cable, Duct Cable, and Microcable. Their cross-sectional structures are illustrated in Figure 1a–c. Each cable type was chosen to represent a different design philosophy in telecommunication infrastructure, ranging from ruggedized, armoured constructions to lightweight, compact configurations.

The Anti-Rodent Cable (Figure 1a) incorporates multiple protective layers to withstand harsh environments and mechanical threats. It is constructed with a black UV-resistant thermoplastic outer jacket and an inner High-Density Polyethylene (HDPE) jacket for added robustness. Beneath these layers, Glass-Reinforced Plastic (GRP) round rod armour provides mechanical reinforcement, while water-blocking tape, jelly filling, and Polybutylene Terephthalate (PBT) loose tubes protect the fibres against moisture ingress. This highly protected structure improves durability but can reduce strain transfer efficiency to the fibre core. Typical applications are surface connections, e.g., in a desert environment or between nearby local buildings (internet).

The Duct Cable (Figure 1b) is designed for underground duct installation, where long-term mechanical reliability is essential. Its structure includes a corrugated steel tape armour beneath the outer sheath, which significantly enhances crush resistance. Loose tubes filled with jelly enclose the fibres, while a central FRP strength member and binding yarn maintain cable integrity. Although the steel armour provides strong mechanical protection, it may also act as a barrier, limiting the extent of strain coupling from the environment to the fibres. The typical use is for Fibre-to-the-home (access to buildings), and global internet infrastructure as the duct allows easier installation and potential swap in the future.

In contrast, the Microcable (Figure 1c) represents a lightweight and compact design, typically deployed where space efficiency and ease of installation are priorities. The fibres (G652D) are arranged in buffer tubes filled with thixotropic jelly for environmental protection. A central FRP strength member and water-blocking yarns provide basic structural support and moisture resistance. Unlike the armoured cables, the Microcable has minimal protective layers, which is expected to enhance strain transfer efficiency and thus improve DAS sensitivity. Environments where one can find Microcables are, amongst others, rail tunnels, data centres, and healthcare centres.

Table 1 summarizes the structural and functional features of the three cables. The comparison highlights how the differences in protective layering, strength members, and filling compounds may influence strain transfer efficiency and, consequently, the DAS response. While armoured cables such as the Anti-Rodent and Duct Cables offer superior durability, their multiple protective layers may attenuate the strain signals. The Microcable, on the other hand, with its simpler construction, is anticipated to yield stronger DAS responses under identical excitation conditions.

The three cables were selected because they represent commonly deployed construction categories in commercial telecommunication networks, spanning lightweight single-jacket indoor cables (Microcable), standard outdoor cables with corrugated steel reinforcement (Duct Cable), and heavily protected multi-layer armoured designs for harsh environments (Anti-Rodent Cable). This selection provides a representative range of structural configurations for evaluating strain-coupling performance.

### 2.2. DAS Interrogator Unit

All distributed acoustic measurements in this study were performed using a NBX 7000 series interrogator supplied by Neubrex Co., Ltd. (Osaka, Japan). The system operates based on the principle of coherent Rayleigh backscattering [15], where laser pulses are launched into the optical fibre where it generates backscattered signals due to the interaction with the glass lattice. These backscatter signals are analyzed to detect dynamic strain variations along the sensing length, often referred to as gauge length. The interrogator converts these variations into time- and distance-resolved amplitude traces, which are proportional to local strain changes within the fibre.

The NBX 7000 series DAS system reports signal amplitude in arbitrary units (a.u.) because the instrument measures the relative change in the Rayleigh backscattered signal rather than converting it into physical strain (με). These a.u. values represent the strength of the optical response produced by the applied vibration: higher values indicate stronger strain transfer to the fibre, while lower values indicate weaker coupling. Although the units are dimensionless, they provide a reliable basis for comparing the relative performance of the three cables because all measurements were carried out under identical interrogation settings and excitation conditions.

The system’s high sensitivity and spatial resolution make it suitable for characterizing strain transfer performance in telecommunication cables used for DAS applications.

### 2.3. Experimental Setup

Each cable sample was tested under identical laboratory conditions to ensure a fair comparison of its distributed acoustic sensing performance. To achieve consistent mechanical coupling along the sensing length, A rigid closed-cell foam block was used as the coupling medium for all three cable types. The density of the foam was not specified by the manufacturer and could not be quantified; however, because the same block and trench configuration were applied uniformly across all samples, the unknown density does not introduce bias between cable types. For each cable, the trench width was machined slightly narrower than the cable diameter (≈0.5–1 mm) to produce a consistent press-fit along the 100 cm sensing region. Although the exact compressive force and local pressure variations could not be measured, the identical preparation method ensured comparable coupling conditions for all cables. Any residual variability is acknowledged as an inherent limitation of foam-based coupling. Both cable ends were firmly secured to prevent axial movement during vibration. No additional weights or adhesives were applied, ensuring consistent coupling across all tests. This arrangement allowed strain-transfer efficiency to be evaluated under controlled and reproducible boundary conditions.

A small motor-driven vibrator was used to introduce periodic mechanical disturbances into the foam block. The excitation waveform applied to the vibrator is shown in Figure 2, which illustrates the 1 Hz square pulse peak-to-peak voltage $\left(V_{pp}\right)$ 10 V, 50% duty cycle was used as the driving signal. The vibrator was equipped with a short hammer-like metal rod extending from its shaft; this rod oscillated forward and backward in response to the applied waveform, striking the foam at +5 V and retracting at –5 V. This action generated a repeatable, low-frequency mechanical impulse that was transferred through the foam and into the embedded cable. The excitation frequency was fixed at 1 Hz to ensure a stable and repeatable mechanical disturbance across all cable samples. At this low frequency, the actuator provides a consistent stroke amplitude with minimal variability, enabling a fair comparison of strain-transfer behaviour under identical boundary conditions. Higher frequencies were not used, as the small motor-based vibrator produces reduced and inconsistent displacement at elevated frequencies, which could introduce non-structural variability and obscure the intrinsic differences among the cable designs.

A real photograph of the complete setup is provided in Figure 3a, showing the foam block, the cable embedded within the groove, the vibrator’s contact point (see Figure 3b), and the lead-in fibre connection to the interrogator. To complement the photograph, a labelled schematic is presented in Figure 3c, indicating the direction of vibration, the hammer-like rod, the cable under test (CUT), the splicing point with the interrogator’s fibre, and the position of the sensing section. Together, Figure 3a–c) provides a clear and reproducible depiction of the physical configuration used in this study.

During each experiment, the NBX 7000 interrogator continuously recorded Rayleigh-based backscatter data over a 15 s acquisition window. The vibration was manually triggered for approximately 5 s (±1 s variation due to manual switching) around the midpoint of the acquisition period (5–10 s), enabling clear separation of baseline, excitation, and post-excitation intervals within the recorded traces. All interrogator settings listed in Table 2 were kept constant across the three cable tests to ensure strict comparability.

A gauge length of 1 m in Table 2 was selected to enhance sensitivity to low-frequency (1 Hz) excitation while maintaining acceptable noise performance, and the 25 cm spatial resolution provided by the interrogator offered sufficient detail to resolve variations along the 1-metre sensing segment. These parameters were kept constant across all tests to ensure comparability

## 3. Results and Discussions

### 3.1. Overview of DAS Signal Responses

Distributed Acoustic Sensing measurements obtained using the NBX 7000 series interrogator revealed clear variations in response amplitude among the tested cable types. Since all measurements were carried out under identical environmental and excitation conditions, the observed differences primarily reflect the strain transfer efficiency determined by each cable’s structural configuration. The signal amplitudes are expressed in arbitrary units (a.u.), and the distance along the x-axis corresponds to the physical position of the sensing fibre embedded within the foam groove.

The NBX7000 recordings were first converted from the native .BIN format into Trend and Single Graphs using the DAS Viewer software V 2.0. To extract the amplitude–distance profile in a reproducible manner, the peak was not taken directly from manually placing a cursor on the graph. Instead, the complete Trend Graph dataset was exported to Excel, where all temporal samples corresponding to the sensing segment were analyzed numerically. The vibration interval was identified from the repeated 1 Hz excitation cycles, and the exact time index of the global maximum response was determined computationally using Excel functions. The full spatial profile (Single Graph) at this precisely identified time index was then extracted, forming the amplitude–distance curve used for comparison. This approach removes subjective selection error and ensures that the spatial peak corresponds exactly to the interrogator’s highest instantaneous response to the applied perturbation.

### 3.2. Anti-Rodent Cable

Figure 4a,b show the DAS response for the Anti-Rodent Cable. The signal demonstrates a relatively low amplitude response within the active vibration region, with a maximum amplitude of approximately 0.0012 *a.u.* along the sensing distance. The response profile exhibits a well-defined peak corresponding to the excitation position, followed by a gradual decay toward the cable’s end. The reduced amplitude response of the Anti-Rodent Cable can be explained by its multi-layered protective structure, which includes a GRP round rod armour, HDPE inner jacket, and water-blocking layers.

These layers, while providing excellent mechanical protection, attenuate strain transmission from the outer environment to the fibre core. As a result, the DAS system detects weaker backscattered amplitude variations, indicating less efficient coupling between the acoustic vibration and the fibre.

Overall, the Anti-Rodent Cable exhibits stable but low-sensitivity DAS behaviour, making it more suitable for applications where mechanical durability is prioritized over fine acoustic detection sensitivity, such as in buried or exposed outdoor installations prone to rodent interference.

### 3.3. Duct Cable

The DAS response of the Duct Cable, shown in Figure 5a,b, exhibits a stronger and more distinct amplitude pattern compared to the Anti-Rodent Cable. The Trend Graph reveals the expected periodic peaks corresponding to the 1 Hz excitation, and the Single Graph (Figure 5b) shows a distinct, sharply localized amplitude maximum centred around 28–30 m, matching the vibration location. The peak amplitude reached approximately 0.0024 a.u., roughly twice that of the Anti-Rodent Cable.

The Duct Cable thus exhibits good strain-transfer efficiency and precise spatial localization, confirming effective coupling between the cable and the surrounding foam. Its construction, incorporating a single corrugated steel layer and central FRP as strength member provide mechanical robustness while maintaining adequate flexibility for strain transmission. The response shape indicates stable and localized strain sensitivity suitable for practical DAS installations.

### 3.4. Microcable

The Microcable demonstrated the strongest DAS response among the three tested samples (Figure 6a,b). The Trend Graph clearly displays six distinct vibration peaks corresponding to the 1 Hz excitation applied during the 6 s disturbance period. The Single Graph shown in Figure 6b reveals a sharp and well-localized amplitude peak centred around the excitation point, with a maximum value of approximately 0.025 a.u., nearly an order of magnitude higher than the Duct Cable.

This high amplitude, combined with minimal background noise, highlights the Microcable’s excellent strain-transfer efficiency. Its lightweight and minimally layered construction facilitates direct coupling between the external perturbation and the optical fibre core, enabling more effective detection of acoustic-induced strain variations. However, the broader spatial response of the Microcable stems from its compliant structure and lower mass, which collectively reduces the acoustic impedance barrier between the foam and the fibre core. This structural characteristic ensures minimal energy reflection and rapid transmission of the induced strain. Consequently, the signal extends over a wider spatial region, maintaining excellent clarity and a high Signal-to-Noise Ratio (SNR). This performance confirms the Microcable’s strong suitability for DAS applications where sensitivity and distributed response are prioritized over rigid spatial confinement.

### 3.5. Comparative Analysis

The comparative analysis of the three telecommunication cables demonstrates that DAS sensitivity is strongly governed by the cable’s internal structure and the efficiency of mechanical coupling between external perturbations and the optical fibre. Since all three samples share similar HDPE-based jacketing and loose-tube materials, differences in their responses primarily reflect structural configuration, and the degree of strain transfer achievable through their respective layer designs.

The results clearly show that additional protective layers, although beneficial for physical robustness, do not necessarily enhance strain transfer. Their effect depends strongly on how compactly the layers are packaged and how firmly the fibre is coupled within the structure. Tightly bonded or minimally layered designs can maintain effective coupling, whereas loosely buffered or heavily layered constructions tend to attenuate strain transmission and reduce DAS sensitivity.

Among the tested samples, the Microcable produced the highest response amplitude (≈0.025 a.u.) and the strongest overall DAS sensitivity. Its simplified, compliant structure enables strain to reach the fibre core more efficiently, resulting in both higher amplitude and a broader spatial response. This wider strain footprint arises because mechanical deformation can propagate along the fibre rather than being confined by rigid outer layers. By contrast, the Duct Cable containing a corrugated steel layer and central FRP showed a moderate response (≈0.0024 a.u.) characterized by a narrower, sharply localized peak. The Anti-Rodent Cable exhibited the lowest response (≈0.0012 a.u.), consistent with its thick, multi-layered protective design, which introduces mechanical isolation and significant damping that limits strain transfer.

These effects are clearly reflected in the amplitude–distance curves shown in Figure 7. The Microcable displays the broadest strain distribution, while the Duct and Anti-Rodent Cables exhibit narrower profiles due to the acoustic impedance mismatch created by their rigid outer layers (e.g., corrugated steel, glass-reinforced plastic). These layers reflect or absorb a large fraction of the incoming disturbance energy, allowing only a small portion to reach the fibre. Their higher structural damping further accelerates attenuation along the cable length, confining the measurable response to a narrow region around the excitation point.

A notable observation is the discrepancy between the amplitudes in the Trend Graph and those extracted from the Single Graph for the Duct and Anti-Rodent Cables, where the Trend Graph values are nearly an order of magnitude higher. This difference arises because the Trend Graph represents a temporally integrated or envelope-type response that accumulates contributions from background noise, reflections, and loose coupling effects over the acquisition period. The Single Graph, in contrast, captures the instantaneous spatial response at a specific excitation cycle, providing a more accurate measure of the true strain amplitude. The larger mismatch in the Duct and Anti-Rodent Cables indicates more pronounced temporal smoothing and energy dissipation within their structures. In the Microcable, however, the close correspondence between the two representations reflects minimal energy loss, confirming its superior and more direct strain transfer.

These trends align with earlier findings on telecommunication cable designs. Freeland et al. (2017) [13] reported that simple single-jacket dielectric cables (e.g., Corning ALTOS^®^ Dielectric) exhibit higher acoustic sensitivity than armoured constructions with corrugated steel layers (e.g., ALTOS^®^ Armored), attributing the reduced sensitivity in armoured designs to mechanical isolation of the fibre. This corresponds directly to our results: the Microcable structurally similar to a single jacket dielectric cable achieves the strongest DAS response, the Duct Cable (with corrugated steel) shows moderate sensitivity, and the Anti-Rodent Cable, with the most protective layering, demonstrates the weakest strain transfer. This agreement reinforces the conclusion that cable architecture and particularly the presence of metallic or multilayer protective components plays a decisive role in DAS strain-coupling efficiency.

### 3.6. Repeatability and Uncertainty Verifications

To evaluate the consistency and reliability of the DAS measurements, each cable was tested in three independent runs under identical excitation and boundary conditions. Figure 8a–c present the spatial amplitude distance responses (Single Graphs) for the Anti-Rodent, Duct, and Microcable, respectively, with each plot showing the three repeated measurements. Small inset Trend Graphs illustrate the corresponding temporal responses for the same runs.

Across all cables, the spatial curves show highly consistent peak amplitudes and similar shapes, demonstrating stable excitation, fibre coupling, and interrogator performance. The only visible variation is a slight lateral shift in peak location between runs. This shift is not caused by cable movement, since the cable remained fixed inside the foam groove throughout all runs, but instead originates from:Manual triggering variations during activation and deactivation of the vibration source, causing small differences in the exact excitation cycle captured.Cycle-to-cycle fluctuation of the motor-driven hammer, which produces small timing differences in impact phase.The time-to-distance mapping in the interrogator, where small temporal offsets appear as small spatial shifts in the reconstructed profile.

Despite these shifts, the peak amplitudes and widths remain consistent, confirming that the strain-transfer characteristics of each cable are reproducible and robust.

To quantify measurement variability, the peak amplitude ($A_{i}$) from each run was extracted, and the standard deviation (*SD*) is calculated:(1)$$SD=\sqrt{\frac{\sum_{i=1}^{3}(A_{i}-\bar{A}{)}^{2}}{n-1}}$$
where $\bar{A}$ is the Grand mean of all runs. Figure 9 presents an error-bar plot using *SD* to illustrate the uncertainty at the peak location for each cable. The error bars are very small relative to the amplitude differences among cable types, indicating that the NBX interrogator introduces minimal run-to-run variability. This confirms that the pronounced differences observed between the three cable constructions arise from their structural strain-transfer behaviour, not from measurement noise or instability.

The repeatability results also provide direct and quantitative verification of the large sensitivity contrast among the three cable types. In the new experimental runs, the Microcable consistently produced the highest spatial peak amplitude, with the strongest run reaching 0.029 a.u., whereas the Anti-Rodent and Duct cables reached only 0.0018 a.u. and 0.00268 a.u., respectively.

These differences exceed one order of magnitude, while the uncertainty ranges obtained from SD are extremely small, confirming that the separation is intrinsic to the cable structures rather than due to measurement noise or variability.

This conclusion is further reinforced by the Trend Graphs: although the excitation peaks in the time domain were approximately equal in all cables—and in some runs even higher in the Anti-Rodent cable the Microcable still converted that same excitation energy into a much stronger spatial strain response. This clearly demonstrates that the Microcable exhibits superior strain-transfer efficiency, meaning it delivers a larger fraction of the applied mechanical energy to the fibre core. The Anti-Rodent and Duct cables, by contrast, dissipate or reflect much of that energy within their multilayer constructions, resulting in significantly weaker strain reaching the fibre.

Taken together, the consistency of repeated measurements, the narrow uncertainty bands, and the large amplitude separation confirm that the Microcable achieves approximately 10× higher DAS sensitivity under identical excitation and boundary conditions.

### 3.7. Temporal SNR Calculation and Interpretation

A quantitative comparison of temporal signal quality was obtained by calculating the SNR directly from the Trend Graph of each run. For each cable, five vibration peaks corresponding to the 1 Hz excitation cycles were extracted, and the baseline noise was measured from the pre-excitation interval.

The SNR for each run was computed using:(2)$$SNR=\frac{\frac{1}{5}\sum_{k=1}^{5}P_{k}}{SD_{\mathrm{baseline}}},$$
where $P_{k}$ is the amplitude of the *k*th excitation peak, $SD_{\mathrm{baseline}}$ is the standard deviation of the 0–5 s baseline region.

The resulting values for all three cable types, across three repeated runs, are summarized in Table 3.

Although the Anti-Rodent cable shows the smallest strain-transfer amplitude, it unexpectedly produced the highest temporal SNR (mean ≈ 111). This arises from the cable’s heavily armoured, multilayer structure, which suppresses small ambient fluctuations and yields an unusually low baseline noise level. Because temporal SNR is determined by the ratio of signal amplitude to baseline noise, even a weak signal may give a high SNR when the noise floor is extremely low.

In contrast, the Microcable, while exhibiting the highest strain-transfer sensitivity in the spatial domain, shows a moderate temporal SNR (mean ≈ 50). Its compliant structure transmits not only the primary excitation but also small background fluctuations, resulting in a slightly higher baseline noise. The Duct Cable lies between these two behaviours.

These results emphasize that:Temporal SNR reflects noise robustness and baseline stability.Spatial peak amplitude reflects strain-transfer efficiency.Both metrics are needed for a complete assessment of DAS performance.

The combined analysis confirms that the Microcable provides the strongest strain coupling, whereas the Anti-Rodent cable offers the quietest baseline.

### 3.8. Discussion Summary

The comparative evaluation of the three telecommunication cables demonstrates that Distributed Acoustic Sensing performance is governed primarily by the internal mechanical structure of the cable and the effectiveness with which external strain is transferred to the fibre core. The Microcable, characterized by its minimal layering and compliant construction, exhibited the highest spatial strain-transfer sensitivity, with peak amplitudes more than an order of magnitude greater than those of the Duct and Anti-Rodent cables. In contrast, cables incorporating corrugated steel layers or multilayer protective elements displayed significantly reduced strain transmission due to increased structural damping, higher acoustic impedance mismatch, and partial mechanical isolation of the fibre.

The temporal SNR analysis complements the spatial findings by revealing the distinct noise behaviour associated with each cable type. The Anti-Rodent cable produced the highest temporal SNR owing to its very low baseline noise, even though its strain-transfer efficiency was the lowest. The Microcable showed moderate SNR but strong strain coupling, while the Duct cable exhibited intermediate behaviour in both metrics. These distinctions highlight the importance of interpreting spatial and temporal responses together, as SNR alone cannot be used as a proxy for sensitivity.

Repeatability tests confirmed that all cables produced stable and consistent responses across repeated runs, with minimal uncertainty relative to the large inter-cable differences. The combination of reproducible spatial profiles, verified uncertainty bounds, and calculated SNR values provides a comprehensive picture of DAS performance across the three cable constructions. These results underscore that cable architecture, particularly buffering configuration, jacket stiffness, and the presence or absence of metallic layers plays a decisive role in determining DAS suitability when repurposing telecommunication fibres for sensing applications.

While this study provides insights into the performance of the distributed sensing cable, several limitations should be noted. First, only a single interrogator model was used, which may influence system-specific performance and limit direct generalization to other interrogators, despite the fundamental Rayleigh backscattering principle. Second, the excitation frequency was restricted to 1 Hz, preventing evaluation of broadband or higher-frequency responses. Although trench dimensions were controlled to ensure tight and uniform cable–foam contact, the compressive force along the interface was not directly measurable. Foam micro-nonuniformities and small machining tolerances may introduce minor variations in local coupling stiffness; however, the repeatability experiments described in the previous subsection demonstrate that such effects did not significantly affect the observed trends across cable types. Finally, the laboratory foam-based coupling does not fully replicate soil or conduit conditions; however, the relative performance ranking among the three cable types is expected to remain unchanged, as strain-transfer efficiency is governed primarily by cable construction rather than the coupling medium. These limitations should be considered when interpreting the results and applying them to practical scenarios.

## 4. Conclusions

This study presented a quantitative and repeatable comparison of Distributed Acoustic Sensing responses in three commercially available telecommunication cables under controlled laboratory excitation. By integrating spatial strain-transfer measurements, temporal SNR analysis, and a dedicated repeatability and uncertainty assessment, the work establishes a clear structure–response–relationship across the tested cable types.

The Microcable consistently demonstrated the strongest strain transfer sensitivity, producing spatial peak amplitudes approximately an order of magnitude higher than those of the Duct and Anti-Rodent cables. The Duct Cable exhibited moderate coupling with well-defined spatial localization, while the Anti-Rodent cable showed the weakest strain transfer despite producing the quietest temporal baseline and highest SNR. These performance differences were shown to be intrinsic to the cable constructions and far greater than the associated measurement uncertainties.

The findings offer practical guidance for selecting existing telecommunication cables for DAS deployment. Lightweight, minimally layered Microcables are well suited for applications requiring high strain sensitivity, such as indoor monitoring, structural health assessments, and laboratory setups. In contrast, armoured or highly protected cables may be beneficial where environmental durability is prioritized, though with reduced sensing performance. Overall, the results provide a valuable reference for understanding how telecommunication cable architecture influences DAS performance and can inform the design and selection of fibre infrastructures for future distributed sensing applications.

## Figures and Tables

**Figure 1 sensors-25-07600-f001:**
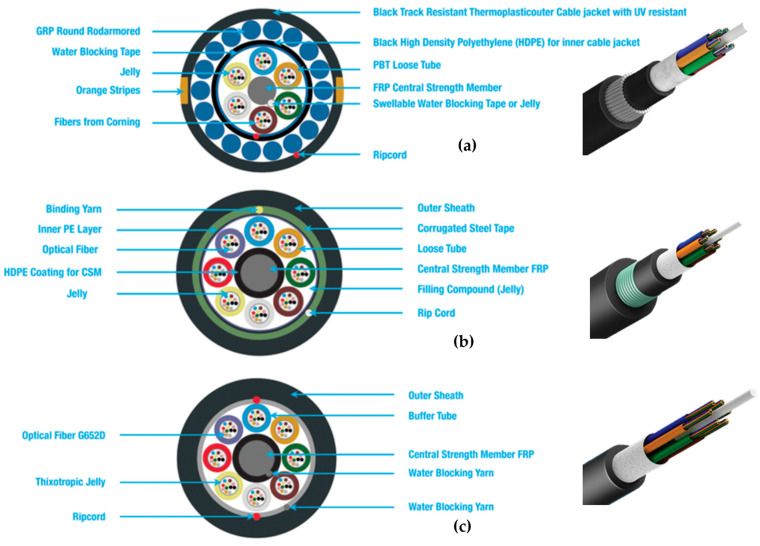
Different telecommunication cables tested for DAS response: (**a**) Anti-Rodent cable, (**b**) Duct cable, and (**c**) Microcable.

**Figure 2 sensors-25-07600-f002:**
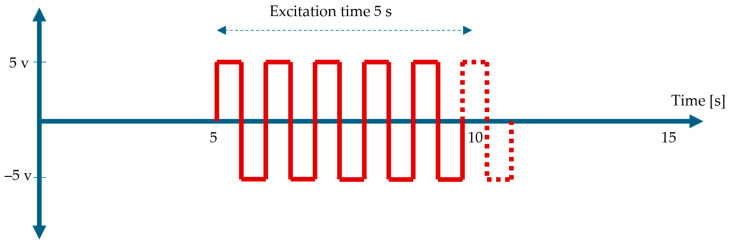
The 1 Hz square-wave excitation signal (±5 V, 50% duty cycle) used to drive the vibrator during the 5–6 s vibration period within the 15 s data acquisition window.

**Figure 3 sensors-25-07600-f003:**
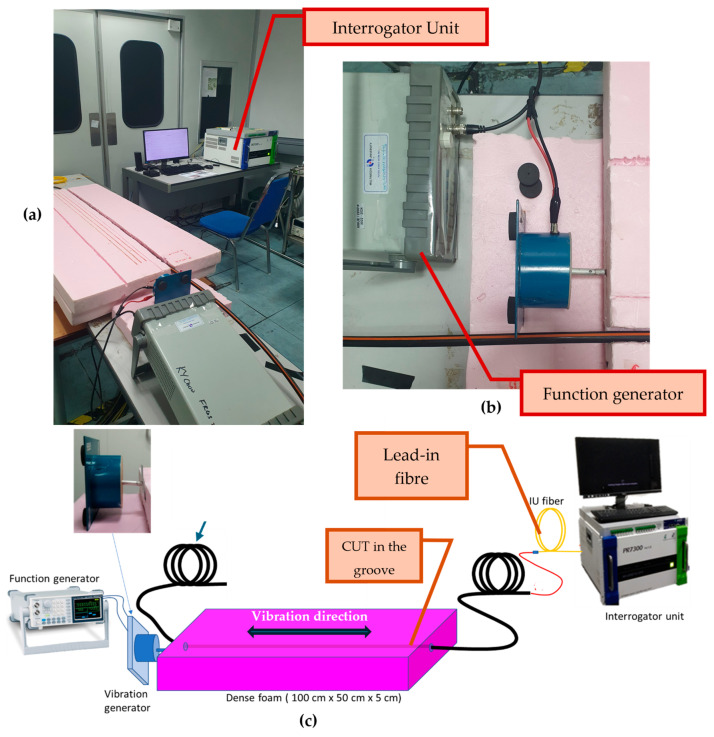
Experimental setup, (**a**) real and (**c**) schematic showing the optical cable embedded in a rigid foam block and excited by a 1 Hz vibrator, with the DAS interrogator (NBX 7000 series, Neubrex Co., Ltd.) used for signal acquisition, and (**b**) top view of the vibrator and actuator rod location relative to the foam and cable.

**Figure 4 sensors-25-07600-f004:**
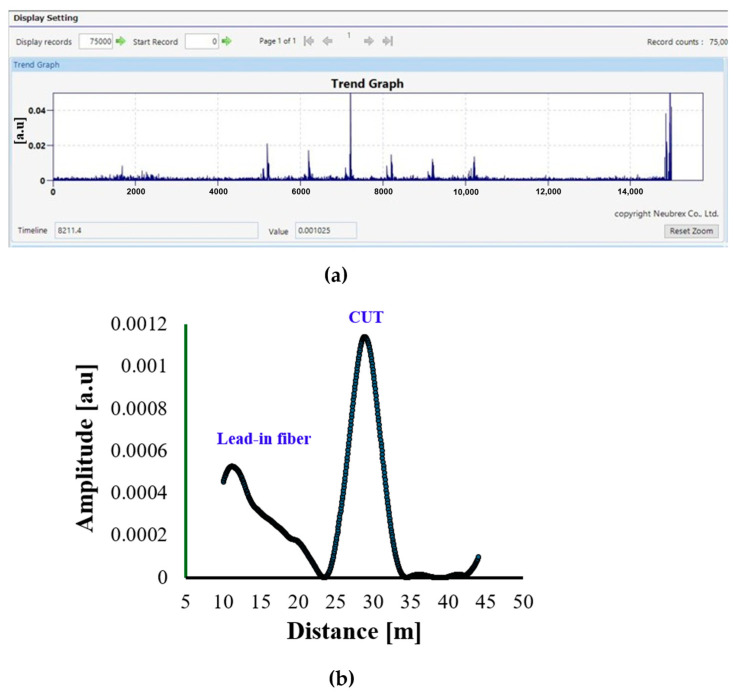
DAS response of the Anti-Rodent Cable: (**a**) Trend Graph showing the temporal variation in signal amplitude during the 15 s acquisition period with 1 Hz excitation applied between 5 and 10 s; (**b**) Single Graph representing the corresponding amplitude–distance profile (in a.u.) extracted from the vibration peaks.

**Figure 5 sensors-25-07600-f005:**
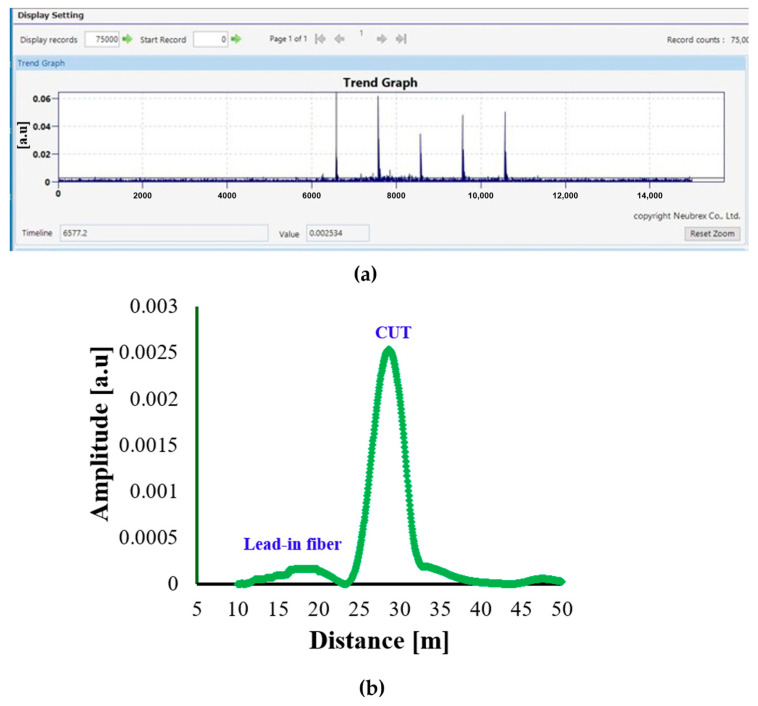
DAS response of the Duct Cable: (**a**) Trend Graph showing periodic peaks associated with the 1 Hz vibration; (**b**) Single Graph showing the spatial amplitude distribution along the sensing distance with a prominent response near the excitation region.

**Figure 6 sensors-25-07600-f006:**
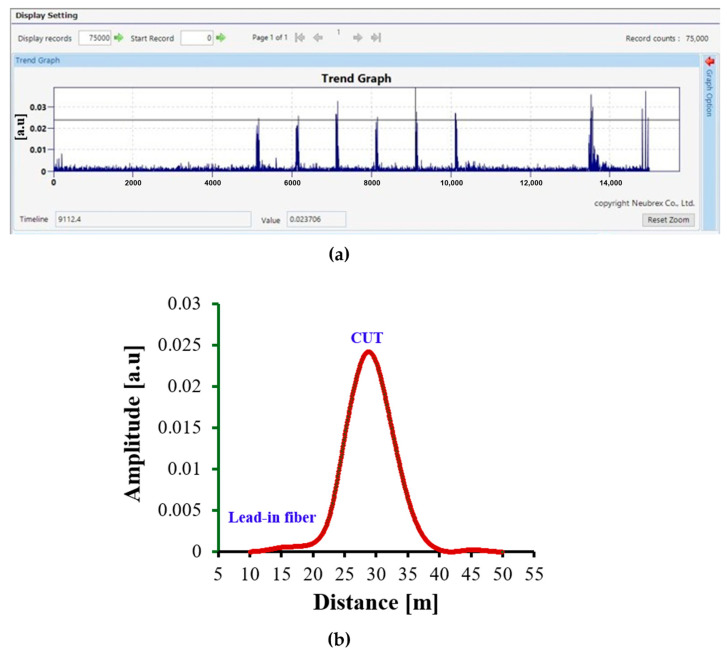
DAS response of the Microcable: (**a**) Trend Graph displaying six distinct vibration peaks corresponding to the 1 Hz excitation window; (**b**) Single Graph showing a sharp and localized amplitude peak near the excitation point, indicating efficient strain transfer.

**Figure 7 sensors-25-07600-f007:**
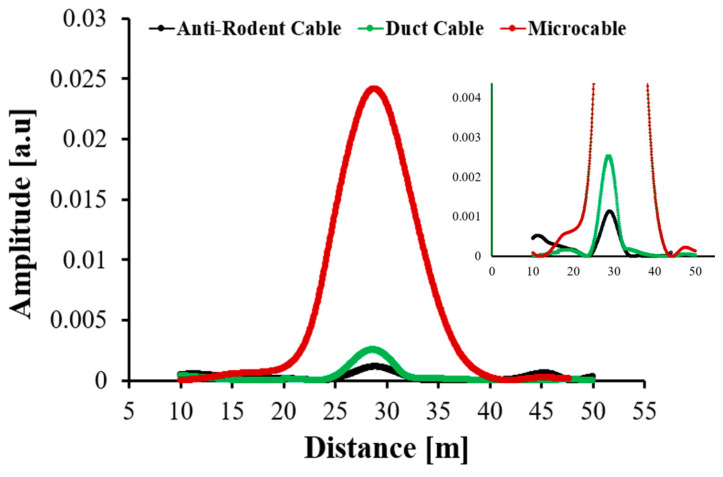
Comparative DAS responses of the three tested telecommunication cables showing amplitude as a function of distance. The inset is focused curves.

**Figure 8 sensors-25-07600-f008:**
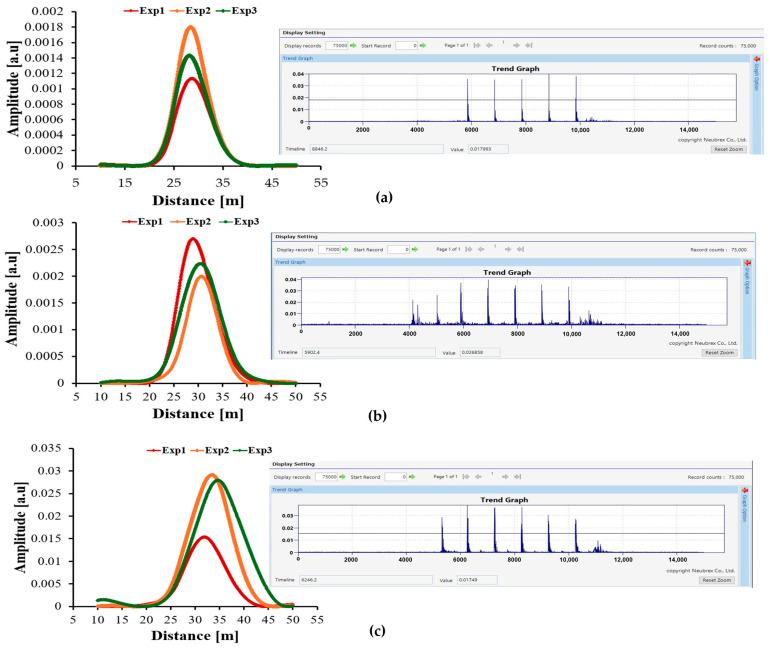
(**a**–**c**) Spatial amplitude–distance responses (Single Graphs) for the Anti-Rodent, Duct, and Microcable cables, respectively, each showing three repeated experimental runs (Exp 1–3). Insets display the corresponding Trend Graph excerpts illustrating the temporal excitation cycles.

**Figure 9 sensors-25-07600-f009:**
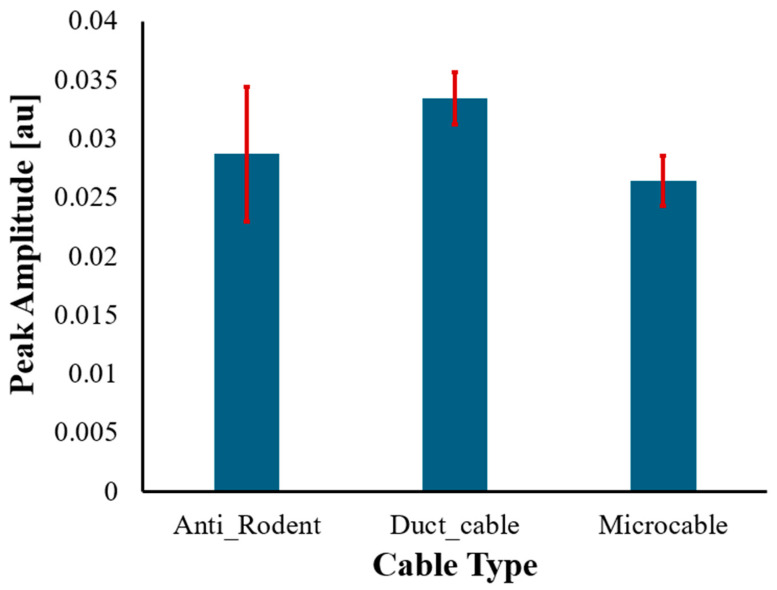
Error-bar plot of the mean temporal peak amplitude for each cable, with standard deviation across the three runs shown as uncertainty.

**Table 1 sensors-25-07600-t001:** Structural and functional features of the three telecommunication optical cables.

Cable Type	Anti Rodent Cable	Duct Cable	Microcable
Outer protection (OD)	UV-resistant thermoplastic outer + HDPE inner (15 mm)	Outer sheath + corrugated steel tape (15 mm)	Outer sheath only (12 mm)
Strength member (OD)	FRP central rod (2 mm) + FRP round armour (1.2 mm)	FRP central rod (2 mm)	FRP central rod (2.8 mm)
Buffer/Tube (OD)	PBT loose tube (2 mm)	PBT loose tube (2 mm)	PBT loose tube (1.5 mm)
Filling/Blocking	Jelly, water blocking tape	Jelly filling compound	Thixotropic jelly, water blocking yarn
Special features	Rodent-resistant, rugged	Duct installation ready, armoured	Lightweight, compact

**Table 2 sensors-25-07600-t002:** DAS interrogator parameters configured in the NBX 7000 series software programme.

Parameters	Value	Remarks
Gauge length	1 m	Determines spatial averaging and affects sensitivity/resolution
Sampling frequency	5 KS/s	Defines temporal resolution of acoustic response
Acquisition duration	15 s	Corresponds to total test period per cable
Excitation duration	5 s ± 1 s (within 15 s window)	Active vibration interval
Frequency range	1 Hz	Set by function generator for the vibrator
Sampling interval	5 cm	-
Spatial resolution	25 cm	-
Output unit	Arbitrary units (a.u.)	Relative amplitude representation

**Table 3 sensors-25-07600-t003:** Signal-to-noise ratio (SNR) computed from the Trend Graph for each cable in each of the three repeated measurements.

Cable	Anti-Rodent	Duct Cable	Microcable
SNR Run1	30.4	25.64	29.78
SNR Run2	273.4	27.59	42.41
SNR Run3	30.28	41.52	79.07
Mean SNR	111.4	31.58	50.42

## Data Availability

The original contributions presented in this study are included in the article. Further inquiries can be directed to the corresponding authors.

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
