# Peer review of "Comparative Study of Distributed Acoustic Sensing Responses in Telecommunication Optical Cables"

_sensors, 2025, doi:10.3390/s25247600_

Round 1

Reviewer 1 Report

Comments and Suggestions for Authors

The paper presents an interesting method to comparing the Distributed Acoustic Sensing (DAS) responses of various telecommunication optical cables. The paper entitled "Comparative Study of Distributed Acoustic Sensing (DAS) Responses in Telecommunication Optical Cables" is interesting and worth disseminating. However, several aspects of the methodology, results presentation, and discussion require significant refinement to meet the standards of a technical publication. A major revision is recommended, with particular attention to the following points:

  1. The paper explains that results are in "arbitrary units," a more thorough explanation of what these units represent physically (e.g., proportional to strain rate, displacement, or optical phase change) would be beneficial. This would help readers understand the physical meaning of the amplitude differences.
  2. The description of the vibrator and its interaction with the foam block could be more specific. A diagram illustrating the exact point of contact, the mechanism of force transfer, and a more precise quantification of the "hammer-like metal rod" movement would enhance reproducibility. Figure 3 is helpful, but it could be improved by adding labels to indicate the lead-in fiber, the cable under test, and the exact location of the vibrator relative to the sensing segment. Arrows showing the direction of vibration would also be beneficial. Also, real image of setup would be more beneficial, how these optical fiber cables distribute on form block.
  3. Resolution of Fig. 1 is too low, needs improvement.
  4. Table 3 provides "Observed SNR," the paper could elaborate more on how SNR was calculated or estimated for each cable type. Given that Microcable has "Very high" SNR, a brief discussion of its factors influencing this would be valuable.
  5. The introduction extensively reviews DAS applications, the discussion section could strengthen its comparative analysis by explicitly referencing how the observed strain transfer efficiencies align with or deviate from findings in other studies on similar cable types or protection schemes.
  6. The selected cables "represent typical constructions." A brief justification for choosing these three specific types (Anti-Rodent, Duct, Microcable) as representative for the comparison would be useful (e.g., market prevalence, distinct structural features).
  7. Table 2 lists gauge length (1 m) and spatial resolution (25 cm), a short explanation of their significance in the context of DAS measurements and how they were optimized for this study would be beneficial.
  8. The paper mentions (pg#12) "temporal smoothing" in the context of the Trend Graph. A more detailed explanation of what this entails and how it affects the perceived amplitude would clarify this point for readers less familiar with DAS signal processing.
  9. In Figure 7, the amplitude responses for Anti-Rodent and Duct cables are very low compared to the Microcable, making it difficult to discern their individual profiles clearly. Consider either using a logarithmic scale for the y-axis or providing an inset with a zoomed-in view for the lower-amplitude cables to show their trends more distinctly.

Reviewer 2 Report

Comments and Suggestions for Authors

Peer Review Report – Sensors (MDPI)

Manuscript ID: sensors-4005327
Title: Comparative Study of Distributed Acoustic Sensing (DAS) Responses in Telecommunication Optical Cables

  1. General Assessment

This manuscript presents an experimental comparison of DAS signal responses in three telecommunication optical cables (Anti-Rodent, Duct, and Microcable) using a Rayleigh-based interrogator under controlled vibration conditions. The topic is relevant and suitable for Sensors, especially given the growing interest in repurposing telecom fibres for distributed sensing applications.

The authors provide a clear motivation and a well-organized experimental workflow. The results are generally consistent with expected strain-transfer behaviour based on cable structure. At this stage, the paper is not ready for publication due to problems of scientific clarity, uncertainty quantification, readability of figures, and improved explanation of the novelty.

  2. Major Comments 

  • a.Novelty and contribution need clearer emphasis.

While the study compares DAS performance in common telecom cables, the novelty is not clearly articulated. Similar work exists regarding strain-transfer efficiency in fibers and cables. The authors should specify:

  • What scientific or practical gap the study fills,
  • How it differs from previous DAS comparative works,
  • Whether this is the first quantitative evaluation of these specific cable constructions.

A dedicated paragraph explaining the unique contribution is needed.

b.Uncertainty and repeatability of measurements.

No uncertainty analysis or repeatability study is reported.

The authors should:

  • Provide error bars or variability for the amplitude values.
  • Discuss SNR quantitatively.
  • indicate whether the measurements were repeated n ≥ 3,
  • Provide uncertainty ranges for the NBX interrogator readings.
  • This is crucial to demonstrate claims like “10× higher sensitivity.”                  c. Methodological clarification of strain extraction

More information is needed about:

  • How the peak amplitude was extracted from the Trend Graph.
  • Whether the highest peak or the average was used,
  • How distance alignment was ensured across the three cables,
  • How tight the cable-foam coupling was (pressure? fixed constraints?).
  • A more precise description of methodology would enhance reproducibility. d. Interpretation of Trend Graph vs Single Graph results.

According to the manuscript, amplitudes in the Trend Graph differ by nearly one order of magnitude from the Single Graph for some cables.

That is to say:

  • Trend Graph integrates temporal energy;
  • Single Graph is instantaneous.

This discrepancy should be explained scientifically, preferably referring to temporal averaging or envelope extraction mechanisms.

  1. Minor Comments

The English is understandable generally, but it needs some grammatical polishing.

  • Define all acronyms at first use; for example, FRP, PBT, HDPE.
  • Explain why the excitation frequency was fixed to 1 Hz; could higher frequencies change relative sensitivity?
  • Captions should briefly explain what the main peak represents; for example, “location of vibration source”.
  • References [6]–[14]: Uniform formatting and missing DOIs fixed.
  • Consider adding a picture of the actual setup in addition to the schematic.
  • Questions for the Authors
  • What is the uncertainty range of the amplitude measurements in the NBX 7000 system?
  • Were the DAS measurements repeated to confirm reproducibility?
  • How exactly was the maximum vibration peak extracted from the Trend Graph?
  • Could differences in cable-foam mechanical contact have influenced results?
  • How would the cable responses change at higher excitation frequencies (>10 Hz)?
  • Can the authors discuss applicability to real field-scale deployments (buried cables, submarine cables)?
  • Is the broader spatial response of the Microcable an artifact of mechanical compliance or interrogator bandwidth?                                    5.Recommendation

This study addresses an important topic for distributed sensing using telecommunication fibres. The experimental design is adequate, and the conclusions are plausible. However, the manuscript requires significant improvement in scientific clarity, uncertainty discussion, and figure presentation.

Round 2

Reviewer 1 Report

Comments and Suggestions for Authors

I recommend this paper for acceptance, subject to minor revisions as follows: The figures are not correctly linked to the text in the manuscript. The authors should carefully review the entire manuscript and ensure that all figures, tables, and equations are correctly referenced. For example: This passage is for Fig. 2 instead of Fig. 3"A real photograph of the complete setup is provided in Figure 3(a), showing the foam block, the cable embedded within the groove, the vibrator’s contact point (see Figure 3(b)), and the lead-in fibre connection to the interrogator. To complement the photograph, a labelled schematic is presented in Figure 3(c),"

Reviewer 2 Report

Comments and Suggestions for Authors

Manuscript ID: Sensors-4005327
Title: Comparative Study of Distributed Acoustic Sensing (DAS) Responses in Telecommunication Optical Cables

  1. Overall Evaluation.

The revised version makes some solid improvements in structure and clarity. It also does better with justifying the experiments. The authors expanded on the methods they used. They included analysis of uncertainties with numbers. Plus they beefed up the comparisons between different cable types. Now the manuscript reads clearer overall. It feels more rigorous too. That puts it closer to what Sensors expects.

Still some key problems linger though. They involve the limits of the setup in the experiments. There are no statistical tests included. Details on calibrating the interrogator are missing. The same goes for how consistent the foam coupling is. A few spots need clearer writing as well.

The work holds real value. It stays technically relevant. But it needs moderate changes before we can accept it.

  1. Strengths of the Manuscript
  • Clear experimental design comparing three telecommunication cables under identical conditions (pages 5–7).
  • Strong use of visuals, including cross-sections and Trend/Single Graphs (Figures 1–6).
  • Meaningful quantitative comparison of strain-transfer efficiency, including peak amplitudes (pages 8–12).
  • New repeatability analysis with uncertainty estimation (page 13–14), which greatly strengthens credibility.
  • Clear practical relevance for DAS deployment in real infrastructures (Conclusion section).
  1. Major Comments

3.1 The influence of foam coupling gets insufficient discussion. This covers pages 5 to 7.

The mechanical interface between the cable and foam plays a critical role. Still, some key aspects lack attention.

  • Foam density remains unspecified in the material.
  • Variability in compressive force along the trench stays unmeasured.
  • Contact uniformity gets assumed without any verification.

Provide more detail about foam characteristics. Or discuss this limitation in greater depth.

3.2 Trend vs. Single Graph discrepancy (page 12)

The manuscript notes that the Trend Graph amplitudes are ~10× higher than Single Graph for certain cables, attributing this to “temporal smoothing.”
This explanation is qualitative.

3.3  Limitations section needed

While the Discussion touches on limitations, a standalone section or paragraph is needed.
Suggested limitations:

  • Only one interrogator model used.
  • Only 1 Hz excitation, no broadband analysis.
  • Laboratory foam-based coupling differs from real deployment.
  1. Minor Comments (Improve Clarity & Quality)
  • English proofreading: several small grammatical issues remain (e.g., page 5, “were assured that firmly fixed”).
  • Figure clarity:
    • Figure 1 is repeated twice; please replace the second occurrence of Figure 1 with Figure 2 on page 6.
  • References:
    • Check formatting of references [14] and [15] (URLs should be accessed-date formatted correctly).
  • Improve Abstract: briefly mention the magnitude of sensitivity differences (e.g., “order of magnitude difference”) to strengthen impact.
    1. Recommendation

The manuscript is promising, technically solid, and close to publishable quality.
The scientific core is correct, but essential clarifications, statistical validation, and a limitations discussion are needed before acceptance.
